# Attachment style and parental bonding: Relationships with fibromyalgia and alexithymia

Annunziata Romeo[1]◯*, Marialaura Di Tella[1]◯, Ada Ghiggia[1], Valentina Tesio[1], Enrico Fusaro[2], Giuliano Carlo Geminiani[1,2], Lorys Castelli[1]

1 Department of Psychology, University of Turin, Turin, Italy, 2 'Città della Salute e della Scienza' Hospital, Turin, Italy

◯ These authors contributed equally to this work.
* annunziata.romeo@unito.it

## Abstract

**Data Availability Statement:** All relevant data are within the paper and its Supporting Information files.

### Objectives

Fibromyalgia (FM) is a chronic pain syndrome, and alexithymia, which is a condition that is characterised by deficits in emotional self-awareness, is highly prevalent among individuals with FM. Insecure attachment styles and inadequate parental care appear to play an important role in the onset and maintenance of both alexithymia and chronic pain. Therefore, the present study aimed to examine the associations between attachment styles, parental bonding, and alexithymia among patients with FM and healthy controls (HC).

### Methods

All participants completed a battery of tests that assessed alexithymia, attachment styles, and parental bonding. Two logistic regression models were tested to examine whether these variables predict (a) group membership (i.e. patients with FM vs. HC) and (b) the likelihood of having alexithymia (i.e. among patients with FM and HC).

### Results

Alexithymia (i.e. difficulty identifying and describing feelings subscales of the 20-item Toronto Alexithymia Scale) significantly predicted group membership (i.e. the likelihood of having FM). On the other hand, educational level and dismissive attachment (i.e. the discomfort with closeness and relationships as secondary subscales of the Attachment Style Questionnaire) were the only significant predictors of the likelihood of having alexithymia.

### Conclusions

These findings highlight both the relevance of alexithymic traits to the definition of FM and centrality of an insecure attachment style to the manifestation of alexithymia.

**Funding:** Lorys Castelli received a grant from the University of Turin (Ricerca scientifica finanziatadall'Università'; Linea A: www.unito.it) and the Cassa di Risparmio di Torino Foundation project titled, 'Componenti psicologiche e psicosomatiche nella sindrome fibromialgica'. The funders had no role in study design, data collection and analysis, decision to publish, or preparation of the manuscript.

**Competing interests:** The authors have declared that no competing interests exist.

## Introduction

Fibromyalgia (FM) is a syndrome that is primarily characterised by chronic and widespread musculoskeletal pain [1, 2], with high incidence among women [3]. The etiopathogenesis of this syndrome is complex and multifactorial, and a series of other conditions such as physical and mental fatigue, disrupted sleep, headaches, irritable bowel syndrome, psychiatric disorders, and cognitive impairments are often associated with chronic pain [4–8].

In recent years, researchers have begun to redirect their attention towards alexithymia, which is a personality trait that is characterised by difficulties in identifying and describing subjective feelings, restricted imaginative processes, and an externally oriented cognitive style [9–11]. Most studies have reported that there is a high prevalence of alexithymia among patients with FM, and the figures range from 48% to 64% [12–14]. With regard to the aetiology of alexithymia, several theoretical models suggest that negative childhood experiences such as traumatic events and inadequate parental bonding may play an important role in the onset of alexithymia [10, 15–17].

Since the '80s, different authors have proposed developmental frameworks of affect regulation that underscore the important regulatory function of caregivers in modulating an infant's emotional states [18–20]. Accordingly, research has also shown that children with insecure attachments, specifically those with an avoidant attachment style, tend to be unable to express negative emotions, especially in highly stressful situations [21, 22].

In more recent years, Fonagy et al. [23] have expanded pre-existing models by delineating the important correlation that exists between the ability of the caregiver to adequately mirror the child's affective states and the child's capacity to effectively represent, tolerate, and regulate affective states.

Several studies have examined the relationship between parental bonding and alexithymia, using both clinical and healthy populations. For instance, a recent meta-analysis [24] showed that there is a negative association between maternal care and alexithymia and a positive association between maternal overprotection and alexithymia among student samples. Relationships between attachment styles and the capacity to represent affective states have also been observed among other non-clinical adult samples. Indeed, several studies have shown that insecure attachment is related to alexithymia [25–28] and that dismissive attachment, in particular, is linked to dysfunctional emotion regulation processing [29, 30].

With regard to clinical populations, the findings of a meta-analysis that was undertaken by Thorbeg et al. [24] highlighted the significant negative association that exists between parental care and alexithymia; however, a significant positive association was found between parental overprotection and alexithymia. Similarly, Gil et al. [31] reported positive associations between ambivalent attachment styles, parental bonding, and alexithymia among patients with somatoform disorders. Particularly, with regard to chronic pain patients, many studies have investigated the relationship between attachment style and chronic pain [32–34] as well as parental bonding and chronic pain [35]. The findings suggest that insecure attachment and inadequate parental care are significantly associated with chronic pain. However, little attention has been paid to the associations between parental bonding, attachment styles, and alexithymia specifically in patients with FM. Among the few studies available that have examined these variables in patients with FM, Gil et al. [36] showed positive associations between alexithymia scores and those that are yielded by both the 'Maternal Abuse' and 'Paternal Indifference' subscales of the Measure of Parental Style (MOPS). Moreover, Peñacoba et al. [37] found positive associations between alexithymia and insecure attachment (both anxious-ambivalent and avoidant attachment styles) in their sample of patients with FM.

The present study aimed to examine deeply the associations between attachment styles, parental bonding, and alexithymia among patients with FM and healthy controls (HC).

Particularly, we aimed to discern if parental bonding and adult attachment styles might play a key role in predicting group membership (i.e. patients with FM vs. HC) or otherwise if these variables could only predict the likelihood of having alexithymia.

## Materials and methods

### Participants and procedure

One hundred female participants with FM were consecutively recruited from the Fibromyalgia Integrated Outpatient Unit (FIOU), which is a multidisciplinary unit that functions based on collaborations between rheumatologists, psychologists, and psychiatrists at the '*Città della Salute e della Scienza*', Turin, Italy. All patients had a primary diagnosis of FM, which had been made by an expert rheumatologist in the field, using ACR criteria of 2010 [2]. The usual clinical practice for patients with FM presenting themselves at our unit includes a first visit with the rheumatologist that made/confirm the diagnosis of FM and a second visit with a psychologist and a psychiatrist together with the rheumatologist in order to formalise the patient care by the FIOU. During a separate session, participants filled out psychological scales after a clinical and psychological interview that assessed sociodemographic and clinical characteristics.

The recruitment took place in the period from September 2016 to January 2018. Patients were recruited consecutively; therefore, the resulting sample is more likely to represent the target population than one resulting from simple convenience sampling.

One hundred and seven healthy women were recruited in order to match the demographic characteristics (i.e. age, gender and educational level) of the FM patients and assigned to the HC group. Healthy women were enrolled from different social and cultural backgrounds in a community sample in Turin. Participants filled in paper-and-pencil versions of the questionnaires, during a face-to-face meeting. The criteria for exclusion from both the FM and HC groups were as follows: being younger than 18 years of age, having a low educational level (< 5 years), lacking fluency in the Italian language, and the presence or a history of a neurological or psychiatric disorder. Furthermore, the presence of rheumatic diseases or chronic pain was included as an additional exclusion criterion for the HC sample only.

This study was approved by the ethics committee of the '*Città della Salute e della Scienza*', Turin, Italy (N. CS/506), and was conducted in accordance with the Declaration of Helsinki. All participants provided written informed consent to participate in the study.

**Measures.**    No previous study has been published yet, using the present dataset.

**Sociodemographic and clinical information.**    Participants were asked to provide sociodemographic (i.e. age, educational level, marital status, and occupation) and clinical information (i.e. duration and severity of illness). Particularly regarding the educational level, we asked both patients with FM and HC to indicate the total of how many years of education they achieved.

In addition, as an index of pain intensity for the patients with FM, the item "Pain" of the Italian version of the Revised Fibromyalgia Impact Questionnaire (FIQ-R) [38, 39] was used to assess the average intensity of pain in the previous week on a scale ranging between 0 and 10.

**Alexithymia.**    The Toronto Alexithymia Scale (TAS-20) [40, 41] is a self-report instrument. It comprises 20 items, each of which requires responses to be recorded on a 5-point Likert-type scale. Item scores yield a total score and three subscale scores. The three subscales assess the different features of alexithymia as follows: (a) difficulty identifying feelings (DIF), which refers to the inability to distinguish between specific emotions or emotions and the bodily sensations of emotional arousal; (b) difficulty describing feelings (DDF), which refers to the inability to verbalise one's emotions to other people; and (c) externally oriented thinking (EOT), which refers to the tendency to direct attention externally rather than towards inner

emotional experiences [41, 42]. The cut-off values and interpretations for the total scores are as follows: ≤ 51 = no alexithymia, 52–60 = borderline alexithymia, ≥ 61 = alexithymia. This scale has demonstrated good internal consistency (Cronbach's α ≥ 0.70) and test-retest reliability [41].

**Parental bonding and attachment style.** The Parental Bonding Instrument (PBI) is a self-report questionnaire that assesses retrospective accounts of parenting styles that an individual had experienced during the first 16 years of life [43, 44]. It consists of 25 items, each of which requires responses to be recorded on 4-point Likert scale. The PBI assesses maternal and paternal parenting styles independently. Consequently, the PBI assesses respondents' perceptions of the relationships that they share with each parent. Two dimensions of parenting styles are measured by the PBI: care and overprotection. A low score on the care subscale is indicative of perceived parental neglect and rejection, whereas a high score is indicative of perceived parental warmth and affection. A high score on the overprotection subscale is indicative of perceived excessive control and intrusive parenting, whereas a low score is indicative of perceived parental acceptance of a child's independence and autonomy. The instrument has demonstrated strong psychometric properties, including long-term temporal stability [45] and high internal consistency (Cronbach's α = 0.74–0.95) [43].

The Attachment Style Questionnaire (ASQ) is a 40-item self-report questionnaire that assesses attachment styles among adults [46, 47]. The respondent is required to rate each item on a 6-point Likert scale, which ranges from 'Totally Disagree' to 'Totally Agree'.

The ASQ consists of five subscales: confidence, preoccupation with relationships, relationships as secondary, discomfort with closeness, and need for approval. Particularly, discomfort with closeness is a theme that is central to Hazan & Shaver's conceptualisation of avoidant attachment [48]. Need for approval refers to the need for acceptance and approval from others and is characteristic of fearful and preoccupied attachment [49]. Preoccupation with relationships entails an anxious and dependent approach to relationships, and it is a core feature of Hazan & Shaver's original conceptualisation of anxious/ambivalent attachment [48]. The dimension, relationships as secondary, is consistent with Bartholomew's concept of dismissive attachment [50]. Finally, confidence (in self and others) is indicative of a secure attachment style. In the present study, a dimensional approach was adopted.

All subscales of the ASQ have demonstrated high internal consistency (Cronbach's α = 0.80) and test-retest reliability over a 10-week period (r = 0.76) [46].

## Statistical analyses

Statistical analyses were conducted using the Statistical Package for Social Sciences (SPSS) version 25.0 (IBM SPSS Statistics for Macintosh, Armonk, NY, USA: IBM Corp.).

Indices of asymmetry and kurtosis were used to test the normality of the data. Values for asymmetry and kurtosis that were between -1 and +1 were considered to be acceptable and indicative of a normal univariate distribution. As per these specifications, all of the variables were found to be normally distributed.

Independent samples $t$-test and Pearson's chi-square test ($\chi^2$) were used to examine group differences in continuous and categorical variables, respectively. Effect size was determined by calculating Cohen's $d$ values.

Finally, two binary logistic regression analyses were conducted. The first logistic regression analysis was conducted to examine if the scores that are yielded by the measures of parental bonding (i.e. PBI–first predictor group entered into the regression model), attachment styles (i.e. ASQ–second predictor group), and alexithymia (i.e. TAS-20 –third predictor group), predict group membership (i.e. participants with FM vs. HC). The second logistic regression

analysis was conducted to examine the effects of demographic variables (first predictor group entered into the regression model), parental bonding (second predictor group), and attachment styles (third predictor group) on the likelihood of having alexithymia. For this analysis, all participants (i.e. both patients with FM and HC) were divided in two groups based on their total scores on the TAS-20 (alexithymic group: total score $\geq$ 61 vs. non-alexithymic group: total score $<$ 61).

To avoid unnecessary reductions in statistical power, only those variables that were significantly different between the two groups (i.e. participants with FM vs. HC or alexithymic vs. non-alexithymic participants), as per the results of preliminary $t$-tests, were included in the logistic regression models. The enter method was used to include the variables of the predictor groups. A $p < .01$ significance level was used to further reduce the likelihood of Type I errors that may result from the conventionally used significance level of $p < .05$. Adjusted odds ratios and 95% confidence intervals were calculated for the predictors of both logistic regression models.

## Results

### Patients with FM versus HC

**Sociodemographic and clinical data.**   The sociodemographic and clinical characteristics of patients with FM and HC are presented in Table 1.

Results of the $t$-tests revealed that patients with FM and HC were matched for age and educational level. With regard to the clinical characteristics of the FM group, patients had had their illness for an average of 8 years and reported a high rate of pain intensity (FIQ-R Pain: 7.56 ± 1.85).

**Alexithymia, parental bonding, and attachment styles.**   Statistics for alexithymia, parental bonding, and attachment styles are presented in Table 2.

With regard to alexithymia, statistical analyses revealed that the FM group obtained significantly higher DIF ($p < .001$, $d = 1.34$) and DDF subscale ($p = .001$, $d = 0.47$) and total ($p < .001$, $d = 0.80$) scores than HC. Participants were classified into categories based on the cut-off

**Table 1.  Sociodemographic and clinical characteristics of the fibromyalgic patients and healthy controls.**  Mean (SD), percentage, and $t$-test are listed.

|  | FM (N = 100) | HC (N = 107) | Test (df) | p |
|---|---|---|---|---|
| Age (years) | 50.15 (10.51) | 47.37 (10.39) | t(205) = 1.910 | .058 |
| Educational level (years) | 11.78 (3.42) | 12.58 (3.01) | t(197.62) = -1.780 | .077 |
| Duration of illness (months) | 97.35 (95.10) | – | | |
| *Marital status* | | | $\chi^2(4) = 9.814$ | **.044** |
| Never-married | 12 (12.1%) | 14 (13.1%) | | |
| Cohabitant | 11 (11.1%) | 11 (10.3%) | | |
| Married | 54 (54.4%) | 72 (67.3%) | | |
| Separated/divorced | 16 (16.2%) | 10 (9.3%) | | |
| Widowed | 6 (6.1%) | 0 (0.0%) | | |
| *Occupation* | | | $\chi^2(4) = 13.470$ | **.009** |
| Student | 3 (3.0%) | 6 (5.6%) | | |
| Employed | 63 (63.0%) | 86 (80.4%) | | |
| Unemployed | 10 (10.0%) | 2 (1.9%) | | |
| Retired | 8 (8.0%) | 6 (5.6%) | | |
| Housewife | 16 (16.0%) | 7 (6.5%) | | |
| FIQ-R Pain | 7.56 (1.85) | – | | |

FM = Fibromyalgia; HC = Healthy Controls; df = Degrees of freedom; FIQ-R = Fibromyalgia Impact Questionnaire Revised version.

**Table 2. Alexithymia, parental bonding, and attachment styles in fibromyalgic patients vs. healthy controls.** Mean (SD), *t*-test, and Cohen's *d* are listed.

| | FM (N = 100) | HC (N = 107) | Test (df) | *p* | Effect size |
|---|---|---|---|---|---|
| *Alexithymia* | | | | | |
| **TAS-20 DIF** | 22.41 (7.27) | 13.56 (5.85) | t(190.045) = 9.611 | **<.001** | *d* = 1.34 |
| **TAS-20 DDF** | 14.10 (5.22) | 11.91 (4.09) | t(187.432) = 3.351 | **.001** | *d* = 0.47 |
| **TAS-20 EOT** | 17.29 (5.02) | 18.27 (4.48) | t(205) = -1.485 | .139 | *d* = 0.21 |
| **TAS-20 Total** | 53.80 (13.72) | 43.74 (11.42) | t(193.080) = 5.713 | **<.001** | *d* = 0.80 |
| *Attachment variables* | | | | | |
| **PBI Maternal Care** | 18.40 (9.06) | 24.19 (7.32) | t(190.416) = -5.057 | **<.001** | *d* = 0.70 |
| **PBI Maternal Overprotection** | 18.46 (8.89) | 13.62 (7.25) | t(191.183) = 4.265 | **<.001** | *d* = 0.60 |
| **PBI Paternal Care** | 17.58 (10.17) | 23.78 (8.15) | t(179.959) = -4.736 | **<.001** | *d* = 0.67 |
| **PBI Paternal Overprotection** | 18.11 (9.54) | 12.90 (8.00) | t(186.244) = 4.168 | **<.001** | *d* = 0.59 |
| **ASQ Confidence** | 29.92 (6.12) | 32.91 (4.67) | t(205) = -3.962 | **<.001** | *d* = 0.55 |
| **ASQ Discomfort with Closeness** | 38.95 (9.01) | 35.06 (7.34) | t(205) = 3.417 | **.001** | *d* = 0.47 |
| **ASQ Relationships as Secondary** | 16.43 (5.87) | 14.86 (5.23) | t(205) = 2.034 | **.043** | *d* = 0.28 |
| **ASQ Need for Approval** | 21.95 (5.79) | 18.62 (5.95) | t(205) = 4.078 | **<.001** | *d* = 0.57 |
| **ASQ Preoccupation with Relationships** | 28.91 (7.83) | 24.79 (6.08) | t(205) = 4.251 | **<.001** | *d* = 0.59 |

FM = Fibromyalgia; HC = Healthy Controls; df = Degrees of freedom; TAS-20 = Twenty-item Toronto Alexithymia Scale; TAS-20 DIF = Difficulty Identifying Feelings factor of Toronto Alexithymia Scale; TAS-20 DDF = Difficulty Describing Feelings factor of Toronto Alexithymia Scale; TAS-20 EOT = Externally-Oriented Thinking factor of Toronto Alexithymia Scale; PBI = Parental Bonding Instrument; ASQ = Attachment Style Questionnaire.

values for the TAS-20 scores. Whereas 35.0% (35/100) of patients with FM were alexithymic and 21.0% (21/100) of them were borderline, 8.4% (9/107) and 15.9% (17/107) of HC were alexithymic and borderline, respectively ($\chi^2$(2) = 26.530, *p* < .001).

With regard to attachment styles, patients with FM obtained lower scores on the confidence subscale of the ASQ, when compared to HC (*p* < .001, *d* = 0.55). However, patients with FM obtained higher scores on the discomfort with closeness (*p* = .001, *d* = 0.47), relationships as secondary (*p* = .043, *d* = 0.28), need for approval (*p* < .001, *d* = 0.57), and preoccupation with relationships (*p* < .001, *d* = 0.59) subscales of the ASQ. Further, with regard to parental bonding, patients with FM obtained higher scores on the maternal (*p* < .001, *d* = 0.60) and paternal (*p* < .001, *d* = 0.59) overprotection subscales of the PBI, when compared to HC. On the contrary, patients with FM obtained lower scores on the maternal (*p* < .001, *d* = 0.70) and paternal (*p* < .001, *d* = 0.67) care subscales of the PBI, when compared to HC.

**Logistic regression.** Hierarchical binomial logistic regression analysis was conducted to examine if alexithymia, parental bonding, and attachment styles predict group membership (i.e. patients with FM vs. HC). Only those variables that were significantly different between the two groups, as per the results of preliminary *t*-tests, were included in the logistic regression models.

In Model 1, the PBI subscale scores were entered as predictors. The model was statistically significant, $\chi^2$ (4) = 39.848, *p* < .001, and the Hosmer-Lemeshow test results were as follows: $\chi^2$ (8) = 3.674, *p* = .885. The model explained 24% (Nagelkerke $R^2$) of the variance and correctly classified 69% of the cases. Among the predictors, both maternal (*p* = .009) and paternal (*p* = .035) care factors were statistically significant (Table 3).

In Model 2, ASQ subscale scores were entered as predictors. The likelihood-ratio test statistic revealed that Model 2 was superior to Model 1 in terms of overall model fit. The block was statistically significant, $\chi^2$ (4) = 18.456, *p* ≤ .001, and the Hosmer-Lemeshow test yielded the following results: $\chi^2$ (8) = 6.142, *p* = .631. The model explained 34% (Nagelkerke $R^2$) of the

**Table 3. Logistic regression predicting likelihood of Fibromyalgia vs. healthy controls based on parental bonding, attachment styles, and alexithymia.**

| | Model 1[a] | | | Model 2[b] | | | Model 3[c] | | |
|---|---|---|---|---|---|---|---|---|---|
| Predictor variables | OR | 95% CI | Wald | OR | 95% CI | Wald | OR | 95% CI | Wald |
| PBI Maternal Care | 1.061 | 1.015–1.109 | 6.746** | 1.062 | 1.012–1.114 | 5.934* | 1.054 | 0.996–1.116 | 3.373 |
| PBI Maternal Overprotection | 0.983 | 0.938–1.029 | 0.552 | 0.989 | 0.940–1.040 | 0.186 | 0.992 | 0.932–1.055 | 0.068 |
| PBI Paternal Care | 1.041 | 1.003–1.081 | 4.449* | 1.028 | 0.987–1.070 | 1.712 | 1.033 | 0.985–1.083 | 1.780 |
| PBI Paternal Overprotection | 0.961 | 0.923–1.002 | 3.483 | 0.961 | 0.918–1.005 | 3.045 | 0.959 | 0.912–1.009 | 2.648 |
| ASQ Confidence | | | | 1.050 | 0.986–1.120 | 2.296 | 1.061 | 0.984–1.144 | 2.344 |
| ASQ Discomfort with Closeness | | | | 0.979 | 0.938–1.022 | 0.957 | 1.005 | 0.956–1.057 | 0.038 |
| ASQ Need for Approval | | | | 0.960 | 0.899–1.024 | 1.534 | 1.038 | 0.958–1.125 | 0.843 |
| ASQ Preoccupation with Relationships | | | | 0.951 | 0.897–1.009 | 2.791 | 0.964 | 0.905–1.027 | 1.279 |
| TAS-20 DIF | | | | | | | 0.772 | 0.706–0.845 | 31.447** |
| TAS-20 DDF | | | | | | | 1.168 | 1.041–1.331 | 6.972** |

[a]$\chi^2$ (4) = 38.848, $p < .001$. Nagelkerke $R^2$ = .24.

[b]$\chi^2$ (8) = 58.304, $p < .001$. Nagelkerke $R^2$ = .34.

[c]$\chi^2$ (10) = 107.980, $p < .001$. Nagelkerke $R^2$ = .56.

OR = Odds Ratio; CI = Confidence Interval; PBI = Parental Bonding Instrument; ASQ = Attachment Style Questionnaire; TAS-20 DIF = Difficulty Identifying Feelings factor of Toronto Alexithymia Scale; TAS-20 DDF = Difficulty Describing Feelings factor of Toronto Alexithymia Scale.

* $p < .05$;

** $p < .01$

variance and correctly classified 72.5% of the cases. Among the predictors, only maternal care ($p = .015$) was statistically significant (Table 3).

In Model 3, the TAS-20 subscale scores were entered as predictors. The likelihood-ratio test statistic revealed that Model 3 was superior to Model 2 in terms of overall model fit. The block was statistically significant, $\chi^2$ (2) = 49.676, $p < .001$, and the Hosmer-Lemeshow test yielded the following results: $\chi^2$ (8) = 3.234, $p = .919$. The model explained 56% (Nagelkerke $R^2$) of the variance and correctly classified 80% of the cases. Maternal care ceased to be statistically significant, and the TAS-20 DIF ($p < .001$) and DDF ($p = .008$) subscales scores emerged as the only two statistically significant predictors (Table 3).

### Alexithymic versus non-alexithymic participants

**Demographic data, parental bonding, and attachment styles.** Statistics for demographic variables, parental bonding, and attachment styles are presented in Table 4.

With regard to attachment styles, alexithymic individuals reported lower scores on the confidence subscale of the ASQ, when compared to non-alexithymic individuals ($p < .035$, $d = 0.34$). Further, the alexithymic group obtained higher scores on the discomfort with closeness ($p < .001$, $d = 0.62$), relationships as secondary ($p < .001$, $d = 0.71$), need for approval ($p < .001$, $d = 0.65$), and preoccupation with relationships ($p < .001$, $d = 0.45$) subscales of the ASQ. With regard to parental bonding, alexithymic individuals obtained lower scores on the maternal ($p = .010$, $d = 0.42$) and paternal ($p < .035$, $d = 0.39$) care subscales of the PBI, when compared to non-alexithymic individuals. There was no significant group difference in maternal and paternal overprotection ($p = $ NS).

**Logistic regression.** Hierarchical binomial logistic regression analysis was conducted to examine the effects of demographic variables, parental bonding, and attachment styles on the likelihood of having alexithymia. Only those variables that were significantly different between

**Table 4. Demographic characteristics, parental bonding, and attachment styles in alexithymic vs. non-alexithymic groups.** Mean (SD), *t*-test, and Cohen's *d* are listed.

| | Alexithymic Group (N = 44) | Non-alexithymic Group (N = 163) | Test (df) | *p* | Effect size |
|---|---|---|---|---|---|
| Age (years) | 51.41 (8.85) | 47.99 (10.83) | t(205) = -1.928 | .055 | d = 0.35 |
| Educational level (years) | 10.59 (2.84) | 12.63 (3.20) | t(205) = 3.826 | **<.001** | d = 0.67 |
| *Attachment variables* | | | | | |
| PBI Maternal Care | 18.39 (9.72) | 22.19 (8.23) | t(204) = 2.612 | **.010** | d = 0.42 |
| PBI Maternal Overprotection | 16.48 (9.31) | 15.83 (8.19) | t(204) = -0.449 | .654 | d = 0.07 |
| PBI Paternal Care | 17.71 (10.92) | 21.68 (9.15) | t(57.100) = 2.161 | **.035** | d = 0.39 |
| PBI Paternal Overprotection | 16.45 (9.45) | 15.07 (9.04) | t(199) = -0.874 | .383 | d = 0.15 |
| ASQ Confidence | 29.89 (6.29) | 31.89 (5.35) | t(205) = 2.120 | **.035** | d = 0.34 |
| ASQ Discomfort with Closeness | 40.93 (8.34) | 35.86 (8.11) | t(205) = -3.659 | **<.001** | d = 0.62 |
| ASQ Relationships as Secondary | 18.91 (6.64) | 14.73 (4.93) | t(54.436) = -3.895 | **<.001** | d = 0.71 |
| ASQ Need for Approval | 23.34 (6.49) | 19.39 (5.72) | t(205) = -3.952 | **<.001** | d = 0.65 |
| ASQ Preoccupation with Relationships | 29.27 (7.00) | 26.10 (7.20) | t(205) = -2.605 | **<.001** | d = 0.45 |

df = Degrees of freedom; PBI = Parental Bonding Instrument; ASQ = Attachment Style Questionnaire.

the two groups, as per the results of preliminary *t*-tests, were included in the logistic regression models.

In Model 1, educational level was entered as a predictor. The model was statistically significant, $\chi^2$ (1) = 14.886, *p* < .001, and the Hosmer-Lemeshow test results were as follows: $\chi^2$ (3) = 4.937, *p* = .176. The model explained 11% (Nagelkerke $R^2$) of the variance and correctly classified 78.2% of the cases. Educational level was found to be a statistically significant predictor (*p* < .001) (Table 5).

In Model 2, maternal care was entered as a predictor. The likelihood-ratio test statistic revealed that Model 2 was superior to Model 1 in terms of overall model fit. The block was statistically significant, $\chi^2$ (1) = 4.645, *p* = .031, and the Hosmer-Lemeshow test results were as follows: $\chi^2$ (7) = 13.407, *p* = .063. The model explained 14% (Nagelkerke $R^2$) of the variance and correctly classified 79.1% of the cases. Both educational level (*p* ≤ .001) and maternal care (*p* = .032) emerged as statistically significant predictors (Table 5). Finally, in Model 3, variables

**Table 5. Logistic regression predicting likelihood of alexithymia vs. non-alexithymia based on demographic characteristics, parental bonding, and attachment styles.**

| | Model 1[a] | | | Model 2[b] | | | Model 3[c] | | |
|---|---|---|---|---|---|---|---|---|---|
| Predictor variables | OR | 95% CI | Wald | OR | 95% CI | Wald | OR | 95% CI | Wald |
| Educational level | 0.805 | 0.716–0.904 | 13.328** | 0.812 | 0.721–0.914 | 11.801** | 0.813 | 0.714–0.927 | 9.636** |
| PBI Maternal Care | | | | 0.957 | 0.920–0.996 | 4.597* | 0.963 | 0.922–1.006 | 2.839 |
| ASQ Discomfort with Closeness | | | | | | | 1.064 | 1.010–1.121 | 5.509* |
| ASQ Relationships as Secondary | | | | | | | 1.094 | 1.018–1.176 | 5.509* |
| ASQ Need for Approval | | | | | | | 1.051 | 0.973–1.135 | 1.611 |
| ASQ Preoccupation with Relationships | | | | | | | 1.011 | 0.951–1.074 | 0.113 |

[a]$\chi^2$ (1) = 14.886, *p* <.001. Nagelkerke $R^2$ = .11.

[b]$\chi^2$ (6) = 19.532, *p* <.001. Nagelkerke $R^2$ = .14.

[c]$\chi^2$ (8) = 44.153, *p* <.001. Nagelkerke $R^2$ = .30.

OR = Odds Ratio; CI = Confidence Interval; PBI = Parental Bonding Instrument; ASQ = Attachment Style Questionnaire.

* *p* <.05;

** *p* <.01

pertaining to attachment styles were included. The likelihood-ratio test statistic revealed that Model 3 was superior to Model 2 in terms of overall model fit. The block was statistically significant, $\chi^2$ (4) = 24.621, $p$ = < .001, and the Hosmer-Lemeshow test results were as follows: $\chi^2$ (8) = 9.333, $p$ = .315. This final model explained 30% (Nagelkerke $R^2$) of the variance and correctly classified 81.1% of the cases. Educational level ($p$ = .002) and scores on the relationships as secondary ($p$ = .015) and discomfort with closeness ($p$ = .019) subscales of the ASQ emerged as statistically significant predictors (Table 5).

## Discussion

The present study aimed to investigate the associations between parental bonding, adult attachment styles, and alexithymia among patients with FM and HC. Previous studies have already been carried out by our research group (e.g., [8; 14; 51–53]), in order to investigate the association between alexithymia and other FM-related variables in this clinical population. However, the present report represents a unique and different contribution, as we have never evaluated before attachment styles and the association with alexithymia among patients with FM and HC. In that way, we contributed to increasing both the body of literature on this topic, which has been sparsely investigated to date, and the knowledge on the psychological aspects, with particular regard to the relationship between alexithymia and attachment in patients with FM compared to HC.

As a first goal of the study, we examined whether parental bonding, adult attachment styles, and alexithymia predict group membership (i.e. patients with FM vs. HC).

The present results, consistently with previous findings [36, 54,55], revealed that levels of alexithymia were higher among patients with FM than HC. On the other hand, with regard to parental bonding, patients with FM obtained lower scores on the maternal and paternal care subscales and higher scores on the maternal and paternal overprotection subscales of the PBI. These findings suggest that, when compared to HC, our sample of patients with FM may have experienced low levels of parental warmth and excessive control during their childhood as a result of the negligent, cold, dismissive, and intrusive parenting styles of both their parents. These findings are consistent with the results of one other study, which investigated the role of parenting styles among patients with FM and found that a high prevalence of maternal abuse and paternal indifference was reported by patients with FM [36].

Further support for the association between adverse parenting styles and chronic illness stems from past findings, which showed significant differences between patients with different chronic medical conditions and HC [35, 56, 57]. For instance, Agostini et al. [56] found that patients with irritable bowel syndrome had reported inadequacies in the parenting styles of their parents as well as personal difficulties in demonstrating warmth, understanding, independence, and individuation from the parental bond. In another study, Agostini et al. [57] found that patients with Crohn's disease perceived their parents' behaviours to be characterised by lower levels of maternal care and higher levels of paternal overprotection, when compared to HC.

With regard to the dimensions of adult attachment, patients with FM obtained higher scores on the discomfort with closeness (corresponds to Hazan & Shaver's conceptualisation of avoidant attachment) and relationships as secondary (corresponds to Bartholomew & Horowitz's conceptualisation of dismissive attachment) subscales of the ASQ, when compared to HC. Moreover, our findings suggest that patients with FM are more likely to seek approval and care from significant others (i.e. need for approval subscale of the ASQ) and be anxious about and dependent upon meaningful relationships (i.e. preoccupation with relationships subscale of the ASQ), when compared to HC. Finally, HC obtained significantly higher scores

on the confidence subscale of the ASQ than patients with FM. This suggests that they have greater self-worth (lovability) and believe that other people are generally accepting and responsive. Our results are in line with a previous study of Peñacoba et al. [37], which found that patients with FM reported significantly higher scores compared to the HC group for both insecure attachment dimensions they evaluated (i.e. anxious-ambivalent attachment style–associated with low self-esteem, higher need of approval, and fear of rejection–and avoidant attachment style–characterised by greater emotional self-sufficiency and greater discomfort in intimacy).

To further examine the specific role that each of the examined factors (i.e. alexithymia, parental bonding, and attachment styles) plays in predicting group membership, hierarchical binomial logistic regression analysis was conducted. Contrary to our expectations, neither parental bonding nor attachment styles significantly predicted group membership (i.e. patients with FM vs. HC). In other words, although patients with FM reported higher levels of insecure attachment and greater difficulties in both maternal and paternal bonding than HC, these aspects do not appear to characterise this clinical population. Indeed, maternal care ceased to be a statistically significant predictor, when alexithymia was introduced into the model. In the final model, only alexithymia (both DIF and DDF factors) was a significant predictor of group membership. These results are attributable to the characteristic features of FM. Indeed, patients with FM typically represent a non-homogeneous clinical sample, and they also differ considerably in their personality traits and attachment styles. Furthermore, individuals with insecure attachment styles demonstrate different developmental trajectories and adopt defence mechanisms other than somatisation, which appears to primarily characterise patients with functional somatic syndromes such as FM.

Based on the above-described results, which show that parental bonding and adult attachment styles do not seem to significantly predict group membership (i.e. the likelihood of having FM), and on the available evidence, which highlights a strong association between alexithymia and insecure attachment, as a second aim of the present study we investigated whether alexithymia, rather than FM per se, could be significantly related with parental bonding and adult attachment styles. In order to test this second hypothesis, we first compared alexithymic and non-alexithymic individuals (considering the whole sample) on attachment variables.

The results showed that alexithymic individuals obtained significantly lower scores on the secure attachment dimension and higher scores on all insecure attachment dimensions, when compared to non-alexithymic individuals. These results concur with the findings of past studies that have highlighted the significant relationship that exists between insecure attachment and alexithymia [19, 26, 27, 58, 59]. For instance, Wearden et al. [59] conducted a study among students and found that the fearful aspect of avoidant attachment was associated with both higher levels of alexithymia and a greater tendency to report different medical symptoms.

With regard to parenting styles, alexithymic individuals obtained lower scores on both the maternal and paternal care subscales of the PBI, when compared to non-alexithymic individuals. This suggests that the former group of participants perceived their mothers and fathers to have not been warm and caring towards them during their childhood. Contrary to the meta-analytic findings of Thorbeg et al. [24], no statistically significant differences in scores on the overprotection subscales of the PBI emerged between alexithymic and non-alexithymic individuals.

In order to examine the associations between attachment and alexithymia more deeply, an additional hierarchical binomial logistic regression analysis was conducted. In the final model, educational level and the relationships as secondary and discomfort with closeness subscales of the ASQ emerged as significant predictors that explained the likelihood of having alexithymia. Contrary to our expectations, parenting styles did not emerge as a significant predictor of alexithymia. Indeed, the significant difference in maternal care, that emerged between the

alexithymic and non-alexithymic individuals during the preliminary comparative analyses, ceased to be significant in the regression model.

With regard to adult attachment, both the relationships as secondary and discomfort with closeness dimensions of the ASQ were found to be significant predictors of alexithymia. These findings appear to be in line with those of Peñacoba et al. [37] who found no significant interactions between 'group' (FM vs HC) and 'attachment style' (secure, avoidant, and anxious-ambivalent) for either of the alexithymia factors. Intragroup comparisons revealed, instead, that both patients with FM and HC showed significant differences in the DDF factor of the TAS-20 between secure and avoidant styles, with higher scores reported for avoidant attachment.

A dismissive attachment style, which is resonant with the relationships as secondary and discomfort with closeness subscales of the ASQ, is considered to be related to experiencing consistently unresponsive caregiving practices during the early years. This can cause individuals to become compulsively self-reliant because they develop a negative view of others and a positive view of the self; consequently, they seek less intimacy from their attachment relationships and frequently suppress and deny their feelings [49] consistent with our results, several previous studies have found that dismissive attachment is linked to dysfunctional emotion regulation processes [29, 30, 60].

The present study also has some limitations. First, since we used self-report questionnaires, participants may have under-reported or exaggerated the severity of their symptoms. Performance-based measures or structured interviews should be employed in addition to traditional self-report measures to overcome this issue. Second, the PBI requires participants to retrospectively evaluate the relationships that they shared with their family members during the first 16 years of life. Thus, memory biases and defence mechanisms that such a measurement strategy may activate could have influenced participant responses. Moreover, the present study adopted a cross-sectional design, which does not permit us to draw concrete conclusions about the causality of the emergent relationships. Therefore, longitudinal studies are needed to investigate the association between parenting styles and alexithymia among patients with FM in greater depth.

## Conclusions

The present study represents one of the few attempts to understand the complex relationships that exist between parental bonding, attachment styles, and alexithymia among patients with FM and HC.

In sum, the main results of our study suggest that alexithymic traits are the main characteristic feature of patients with FM. Indeed, although patients with FM primarily reported dysfunctional parental bonding and an insecure attachment style, these two factors do not seem to play a specific role in predicting the likelihood of having FM. Conversely, the two dimensions of insecure attachment (i.e. discomfort with closeness and relationships as secondary; ASQ) appear to play an important role in predicting the likelihood of having alexithymia.

The present findings have important implications for clinical practice. First, when working with patients with FM, clinical attention should be paid to not only the management of pain symptoms but also impairments in affect regulation and attachment dynamics. The use of group therapeutic interventions like the Attachment-Based Compassion Therapy that includes formal practices of mindfulness and visualizations based on self-compassion and the attachment style that was generated in childhood, could be an effective and also cheaper strategy for the treatment of patients with FM [61].

Second, among individuals with high levels of alexithymia, attention should be paid to the establishment of a secure therapeutic alliance because dismissive attachment patterns appear to play a role in the onset and maintenance of alexithymic traits.

## Supporting information

**S1 Dataset. Raw data used in the current study.**
(XLSX)

## Acknowledgments

The authors would like to thank the participants involved in the study.

## Author Contributions

**Conceptualization:** Annunziata Romeo, Marialaura Di Tella, Lorys Castelli.

**Data curation:** Ada Ghiggia, Valentina Tesio.

**Formal analysis:** Annunziata Romeo, Marialaura Di Tella.

**Methodology:** Annunziata Romeo, Marialaura Di Tella, Lorys Castelli.

**Supervision:** Enrico Fusaro, Giuliano Carlo Geminiani.

**Writing – original draft:** Annunziata Romeo, Marialaura Di Tella.

**Writing – review & editing:** Annunziata Romeo, Marialaura Di Tella, Lorys Castelli.

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
