## [Editor Report · Decision Letter 0]

31 Jan 2020

PONE-D-20-01350

Attachment style and parental bonding: relationships with fibromyalgia and alexithymia

PLOS ONE

Dear Dr Romeo,

Thank you for submitting your manuscript to PLOS ONE. After careful consideration, we feel that it has merit but does not fully meet PLOS ONE’s publication criteria as it currently stands. Therefore, we invite you to submit a revised version of the manuscript that addresses the points raised during the review process.

Plos One internal editors have carefully examined your manuscript. They recommend a Major Revision before sending the manuscript out for review. Please resubmit your manuscript addressing the following points:

1. Please provide additional information about the participant recruitment method and the demographic details of your participants. Please add in your methods section such details as a) the recruitment date range (month and year), b) a statement as to whether your sample can be considered representative of a larger population, c) a description of how participants were recruited, and d) descriptions of where participants were recruited and where the research took place.

2. Internal editors noticed that the present submission is closely related to previously published works, including:

- Tella MD, Enrici I, Castelli L, Colonna F, Fusaro E, Ghiggia A, Romeo A, Tesio V, Adenzato M. Alexithymia, not fibromyalgia, predicts the attribution of pain to anger-related facial expressions. J Affect Disord. 2018; 227: 272-279.

- Di Tella M, Tesio V, Ghiggia A, Romeo A, Colonna F, Fusaro E, Geminiani GC, Bruzzone M, Torta R, Castelli L. Coping strategies and perceived social support in fibromyalgia syndrome: Relationship with alexithymia. Scand J Psychol. 2018; 59(2): 167-176.

- Ghiggia A, Romeo A, Tesio V, Tella MD, Colonna F, Geminiani GC, Fusaro E, Castelli L. Alexithymia and depression in patients with fibromyalgia: When the whole is greater than the sum of its parts. Psychiatry Res. 2017 Sep;255:195-197.

- Di Tella M, Ghiggia A, Tesio V, Romeo A, Colonna F, Fusaro E, Torta R, Castelli L. Pain experience in Fibromyalgia Syndrome: The role of alexithymia and psychological distress. J Affect Disord. 2017 Jan 15;208:87-93.

- Tesio V, Di Tella M, Ghiggia A, Romeo A, Colonna F, Fusaro E, Geminiani GC, Castelli L. Alexithymia and Depression Affect Quality of Life in Patients With Chronic Pain: A Study on 205 Patients With Fibromyalgia. Front Psychol. 2018; 9: 442.

Please ensure that the related works have been adequately discussed, by commenting both in your cover letter and in the manuscript on how this work constitutes a distinct contribution. Please also ensure that any overlap in the dataset and analyses with previous works has not compromised the robustness of the statistical analysis (e.g., by removing required adjustments for multiple hypothesis testing). For further information on our submission guidelines on related manuscripts, please see " ext-link-type="uri" xlink:type="simple">http://journals.plos.org/plosone/s/submission-guidelines#loc-related-manuscripts.",

I would like to thank you very much for forwarding your manuscript to us for consideration.

We would appreciate receiving your revised manuscript by Mar 16 2020 11:59PM. To enhance the reproducibility of your results, we recommend that if applicable you deposit your laboratory protocols in protocols.io, where a protocol can be assigned its own identifier (DOI) such that it can be cited independently in the future. For instructions see: http://journals.plos.org/plosone/s/submission-guidelines#loc-laboratory-protocols

We look forward to receiving your revised manuscript.

Kind regards,

Juan V. Luciano, PhD

Academic Editor

PLOS ONE

Journal Requirements:

„The funders had no role in study design, data collection and analysis, decision to publish, or preparation of the manuscript.”

a)    Please provide an amended Funding Statement that declares *all* the funding or sources of support received during this specific study (whether external or internal to your organization) as detailed online in our guide for authors at http://journals.plos.org/plosone/s/submit-now.  

b)    Please state what role the funders took in the study.  If any authors received a salary from any of your funders, please state which authors and which funder. If the funders had no role, please state: "The funders had no role in study design, data collection and analysis, decision to publish, or preparation of the manuscript."

---

## [Author Response · Author response to Decision Letter 0]

24 Feb 2020

Dear Editor, 

please find below a point by point response to all your comments. Thank you for the opportunity to clarify the issues that you raised. 

1. Please provide additional information about the participant recruitment method and the demographic details of your participants. Please add in your methods section such details as a) the recruitment date range (month and year), b) a statement as to whether your sample can be considered representative of a larger population, c) a description of how participants were recruited, and d) descriptions of where participants were recruited and where the research took place.

Following your indications, we added the required information. Specifically, we added the recruitment date range, the recruitment details for both FM patients and HC. Our FM patients were recruited consecutively, so we added the following sentence in the text: “Patients were recruited consecutively; therefore, the resulting sample is more likely to represent the target population than one resulting from simple convenience sampling.”

2. Internal editors noticed that the present submission is closely related to previously published works, including:

- Di Tella MD, Enrici I, Castelli L, Colonna F, Fusaro E, Ghiggia A, Romeo A, Tesio V, Adenzato M. Alexithymia, not fibromyalgia, predicts the attribution of pain to anger-related facial expressions. J Affect Disord. 2018; 227: 272-279.

- Di Tella M, Tesio V, Ghiggia A, Romeo A, Colonna F, Fusaro E, Geminiani GC, Bruzzone M, Torta R, Castelli L. Coping strategies and perceived social support in fibromyalgia syndrome: Relationship with alexithymia. Scand J Psychol. 2018; 59(2): 167-176.

- Ghiggia A, Romeo A, Tesio V, Tella MD, Colonna F, Geminiani GC, Fusaro E, Castelli L. Alexithymia and depression in patients with fibromyalgia: When the whole is greater than the sum of its parts. Psychiatry Res. 2017 Sep;255:195-197.

- Di Tella M, Ghiggia A, Tesio V, Romeo A, Colonna F, Fusaro E, Torta R, Castelli L. Pain experience in Fibromyalgia Syndrome: The role of alexithymia and psychological distress. J Affect Disord. 2017 Jan 15;208:87-93.

- Tesio V, Di Tella M, Ghiggia A, Romeo A, Colonna F, Fusaro E, Geminiani GC, Castelli L. Alexithymia and Depression Affect Quality of Life in Patients With Chronic Pain: A Study on 205 Patients With Fibromyalgia. Front Psychol. 2018; 9: 442.

Please ensure that the related works have been adequately discussed, by commenting both in your cover letter and in the manuscript on how this work constitutes a distinct contribution. 

Please also ensure that any overlap in the dataset and analyses with previous works has not compromised the robustness of the statistical analysis (e.g., by removing required adjustments for multiple hypothesis testing). For further information on our submission guidelines on related manuscripts, please see http://journals.plos.org/plosone/s/submission-guidelines#loc-related-manuscripts.

Following your remarks, we discussed the specificity of this paper with respect to our previous studies on FM patients both in the manuscript (discussion section) and in the opening lines of the cover letter.

Furthermore, we specified that the current dataset has not been used for other studies (we added the following sentence in the text: “No previous study has been published yet, using the present dataset”). We carried out different studies investigating the psychological components of FM patients, but for each study we enrolled different patients in different periods of time.

---

## [Decision Letter · Decision Letter 1]

20 Mar 2020

**PONE-D-20-01350R1**

Attachment style and parental bonding: relationships with fibromyalgia and alexithymia

PLOS ONE

Dear **Dr Romeo**,

Thank you for submitting your manuscript to PLOS ONE. After careful consideration, we feel that it has merit but does not fully meet PLOS ONE’s publication criteria as it currently stands. Therefore, we invite you to submit a revised version of the manuscript that addresses the points raised during the review process.

We would appreciate receiving your revised manuscript by May 04 2020 11:59PM. To enhance the reproducibility of your results, we recommend that if applicable you deposit your laboratory protocols in protocols.io, where a protocol can be assigned its own identifier (DOI) such that it can be cited independently in the future. For instructions see: http://journals.plos.org/plosone/s/submission-guidelines#loc-laboratory-protocols

We look forward to receiving your revised manuscript.

Kind regards,

Juan V. Luciano, Ph.D.

Academic Editor

PLOS ONE

Reviewers' comments:

Reviewer's Responses to Questions

**Comments to the Author**

1. If the authors have adequately addressed your comments raised in a previous round of review and you feel that this manuscript is now acceptable for publication, you may indicate that here to bypass the “Comments to the Author” section, enter your conflict of interest statement in the “Confidential to Editor” section, and submit your "Accept" recommendation.

Reviewer #2: (No Response)

2. Is the manuscript technically sound, and do the data support the conclusions?

Reviewer #1: Partly

Reviewer #2: Partly

3. Has the statistical analysis been performed appropriately and rigorously? 

Reviewer #1: Yes

Reviewer #2: Yes

4. Have the authors made all data underlying the findings in their manuscript fully available?

Reviewer #1: Yes

Reviewer #2: Yes

5. Is the manuscript presented in an intelligible fashion and written in standard English?

Reviewer #1: Yes

Reviewer #2: Yes

6. Review Comments to the Author

Reviewer #1: Introduction.

Two recently published studies reported a higher prevalence of alexithymia in people with Fibromyalgia (48%-64%) than those reported in the introduction. It would be convenient to update this information.

References:

1. Front Psychol. 2019 Jul 31; 10: 1735. DOI: 10.3389/fpsyg.2019.01735.

2. Clin Exp Rheumatol 2019 Oct 9[Online ahead of print]

Methods.

A medical expert in the field made the diagnosis of Fibromyalgia. However, it is not clear if the doctor used any classification criteria (for example, ACR criteria 1990, 2010 or 2016) to unify the standards in the diagnosis in the sample. I consider it appropriate to clarify in the methods section.

Something to complement the description of the sample is if the center where the investigation was carried out, any patient with pain can go to the center (Primary Healthcare), or for their attention, the evaluation and referral of another doctor are necessary (Secondary / Tertiary Health Care).

There is duplicate information; for example, page 5, lines 106 and 112, repeat the time of inclusion of patients. The exclusion criteria are the same for both groups, with the exception that in the control group, patients with rheumatic diseases or chronic pain are excluded. I recommend presenting them in the same paragraph.

Results.

In the manuscript, there is any inconsistency between the number of people with fibromyalgia included. In the methods section, one hundred and seven patients report. In the headings of Table 1 and Table 2, 100 patients are reported. However, the sum of the variable's marital status and occupation includes only 99 patients. This information should be reviewed.

There is information that suggests gender differences in the prevalence of alexithymia. Because of the higher frequency of fibromyalgia in women, some studies only include women. The manuscript does not present the percentage of women involved in each of the groups. This information should be clarified.

Page 11, line 220. Consider changing the word "composite" to "total score" or "global score." Which better reflects the score of the 20 items on the TAS-20 scale.

Conclusions.

In the conclusions, I suggest removing "In spite of these limitations"

Reviewer #2: Attachment style and parental bonding: relationships with fibromyalgia and alexithymia

This article focuses on a particular area of interest, specifically “to examine the associations between attachment styles, parental bonding, and alexithymia among patients with fibromyalgia (FM) and healthy controls (HC)” (lines 22-24), nevertheless the method employed makes it look as if it were two independent studies with no apparent link, as if they weren’t connected enough to be part of the same manuscript: a) to analyze predictive variables for fibromyalgia, b) to analyze predictive variables for alexithymia.

In particular, the manuscript presents two clearly differentiated analyses. One of them, directly associated to the aim of the manuscript, focuses on the differences between alexithymia, attachment, parental bonding and anxiety and depression in FM and HC by means of bivariate analyses and logistic regression. The other, with no association with the main aim, but having used the same method, uses the same analyses, in this case to analyze the differences between participants with and without alexithymia (with the total sample). There is a need to justify this analysis within the general aim of the study. In this context, and as the authors have pointed out (“the present report represents a unique and different contribution, as we have never evaluated before attachment styles and the association with alexithymia in patients with FM” (lines 349-351), the novelty of the manuscript is lost as it actually tackles two independent aims that have been widely reported in previous literature.

The methods should have been adapted, in particular the statistical analyses, so as to put both together (a and b). One possibility could be to include the condition (FM/HC) as a predictive or moderating variable (depending on which aim is being pursued) in the second regression analysis (alexithymia vs no alexithymia).

The following suggestions and modifications should also be considered:

1) To delete the phrase “No previous study has been published yet, using the present dataset” from the measures section (line 123-124).

2) It would be of interest to specify the Cronbach’s alphas obtained from the sample.

3) There is no justification for the inclusion of the anxiety and depression measures as part of the study. The justification hasn’t been sufficiently well argued: “we decided to include also psychological distress variables into the model, considering the prominent role that these factors play in the symptomatology of patients with FM” (lines 197-198). Furthermore, the HADS was used as a measure for psychological distress, and it was later included as a measure for anxiety and depression.

4) There is little clarity regarding the measure used to assess the sample’s educational level. Table 1 states “Educational level (years)”. Given that this is not a usual measure for educational level, what it actually means should be specified in the instruments section.

5) Why was there no suggestion of a sample of healthy participants to match the fibromyalgia patients for age and educational level so to be able to ensure more homogeneity for these variables? Although no statistically significant differences were found, they are close to significance.

6) Were there any statistically significant differences in relation to marriage status or educational level between FM and HC? Chi-square analyses and p. values are missing in Table 1.

7) Please indicate in the text (lines 231-234) the contrast statistic (chi-square) and the p. values for the differences in the proportion of alexithymia between the FM and HC groups.

8) In the discussion, there is no need to comment on the association between parenting styles and other chronic diseases which are not fibromyalgia (lines 367-374). There should be further discussion regarding the results in relation to fibromyalgia populations.

9) In relation to the comment on lines 392-393 “Contrary to our expectations, neither parental bonding nor attachment styles significantly predicted group membership (i.e. patients with FM vs. HC)”, further discussion of this result is required. It is possible that having introduced symptoms (anxiety and depression) as predictive variables could have concealed the role that other variables of interest might have played. The method should be adapted, in particular the statistical analyses, for this limitation, and should include the use of symptom variables as possible modulators and not as predictors.

10) In Model 2 (line 272) “In Model 2, the TAS-20 total and subscale scores were entered as predictors”, only dimensions and not total TAS-20 scores should have been included so to avoid co-linearity problems that could affect results. In fact, one of the dimensions included in the total TAS-20 (“externally oriented thinking”) has not been shown to be an explanatory variable in this study, nor in any other previous studies either.

7. PLOS authors have the option to publish the peer review history of their article (what does this mean?). If published, this will include your full peer review and any attached files.

Reviewer #1: No

Reviewer #2: No

---

## [Author Response · Author response to Decision Letter 1]

25 Mar 2020

Turin, 25 March 2020

Ref: PONE-D-20-01350R1

Title: Attachment style and parental bonding: relationships with fibromyalgia and alexithymia

Journal: PLOS ONE

Dear Dr. Juan V. Luciano,

many thanks for the opportunity to resubmit the above manuscript to “PlOS ONE” journal. We are very grateful for the thorough, insightful reviews and we have modified the paper taking into account all these suggestions. Please, you can find below a point-by-point response to all the reviewers’ comments, in Italic the referee comments; in bold the authors’ responses. Modifications in the text have been highlighted in yellow. 

Waiting for your gentle reply,

Yours Sincerely 

Annunziata Romeo

 

Editor’s Comments

Dear Dr Romeo,

Thank you for submitting your manuscript to PLOS ONE. After careful consideration, we feel that it has merit but does not fully meet PLOS ONE’s publication criteria as it currently stands. Therefore, we invite you to submit a revised version of the manuscript that addresses the points raised during the review process.

Thank you for the opportunity to consider our manuscript for publication.

Reviewers’ Comments to Authors

Reviewer 1

Introduction.

Two recently published studies reported a higher prevalence of alexithymia in people with Fibromyalgia (48%-64%) than those reported in the introduction. It would be convenient to update this information.

References:

1. Front Psychol. 2019 Jul 31; 10: 1735. DOI: 10.3389/fpsyg.2019.01735.

2. Clin Exp Rheumatol 2019 Oct 9[Online ahead of print]

Thank you for your comments. Following your suggestions, we updated the prevalence of alexithymia in patients with Fibromyalgia (FM), according to this new evidence. 

Methods

1) A medical expert in the field made the diagnosis of Fibromyalgia. However, it is not clear if the doctor used any classification criteria (for example, ACR criteria 1990, 2010 or 2016) to unify the standards in the diagnosis in the sample. I consider it appropriate to clarify in the methods section.

Thank you for your comment. For the diagnosis of FM, the rheumatologists used the ACR criteria of 2010. Following your suggestion, we added this information in the ‘Participants and procedure’ section. 

2) Something to complement the description of the sample is if the center where the investigation was carried out, any patient with pain can go to the center (Primary Healthcare), or for their attention, the evaluation and referral of another doctor are necessary (Secondary / Tertiary Health Care).

Thank you for your observation. The usual practice for our center requires that firstly the general practitioner refers the patients to the rheumatologist, so that he/she can ascertain whether the diagnosis of FM may be done. After the diagnosis of FM has been confirmed, the rheumatologist usually refers the patients to the psychiatrist and psychologist, who in turn provide patients with the pharmacological treatment and the psychological support. 

3) There is duplicate information; for example, page 5, lines 106 and 112, repeat the time of inclusion of patients. The exclusion criteria are the same for both groups, with the exception that in the control group, patients with rheumatic diseases or chronic pain are excluded. I recommend presenting them in the same paragraph.

Thank you for your comments. Following your suggestions, we made the required changes to the ‘Participants and procedure’ section. 

Results

1) In the manuscript, there is any inconsistency between the number of people with fibromyalgia included. In the methods section, one hundred and seven patients report. In the headings of Table 1 and Table 2, 100 patients are reported. However, the sum of the variable's marital status and occupation includes only 99 patients. This information should be reviewed.

We appreciate this remark being pointed out to us. A total sample of 100 patients with FM was recruited for the present study. However, regarding the marital status, there has been a missing value and so the sum is 99 patients. For what concern, instead, both the ‘Participants and procedure’ section and the occupation, an error has occurred during the transcription of the data. We have now corrected the mistake in both parts. 

2) There is information that suggests gender differences in the prevalence of alexithymia. Because of the higher frequency of fibromyalgia in women, some studies only include women. The manuscript does not present the percentage of women involved in each of the groups. This information should be clarified.

Thank you for your observation. As indicated in the ‘Participants and procedure’ section, both the FM and HC groups consist entirely of women. Therefore, no percentage for gender has been reported. 

3) Page 11, line 220. Consider changing the word "composite" to "total score" or "global score." Which better reflects the score of the 20 items on the TAS-20 scale.

Thank you for your comment. Following your suggestion, we changed the word ‘composite’ to ‘total score’ for the TAS-20. 

Conclusions

In the conclusions, I suggest removing "In spite of these limitations".

Thank you for your observation. Following your suggestion, we removed ‘In spite of these limitations’ at the beginning of the conclusions. 

Reviewer: 2

Comments to the Author

This article focuses on a particular area of interest, specifically “to examine the associations between attachment styles, parental bonding, and alexithymia among patients with fibromyalgia (FM) and healthy controls (HC)” (lines 22-24), nevertheless the method employed makes it look as if it were two independent studies with no apparent link, as if they weren’t connected enough to be part of the same manuscript: a) to analyze predictive variables for fibromyalgia, b) to analyze predictive variables for alexithymia.

In particular, the manuscript presents two clearly differentiated analyses. One of them, directly associated to the aim of the manuscript, focuses on the differences between alexithymia, attachment, parental bonding and anxiety and depression in FM and HC by means of bivariate analyses and logistic regression. The other, with no association with the main aim, but having used the same method, uses the same analyses, in this case to analyze the differences between participants with and without alexithymia (with the total sample). There is a need to justify this analysis within the general aim of the study. In this context, and as the authors have pointed out (“the present report represents a unique and different contribution, as we have never evaluated before attachment styles and the association with alexithymia in patients with FM” (lines 349-351), the novelty of the manuscript is lost as it actually tackles two independent aims that have been widely reported in previous literature.

The methods should have been adapted, in particular the statistical analyses, so as to put both together (a and b). One possibility could be to include the condition (FM/HC) as a predictive or moderating variable (depending on which aim is being pursued) in the second regression analysis (alexithymia vs no alexithymia).

Thank you for your comments and appreciation. 

The present study had a twofold aim, as we reported in the following lines of the introductive section: “The present study aimed to examine deeply the associations between attachment styles, parental bonding, and alexithymia among patients with FM and healthy controls (HC). Particularly, we aimed to discern if parental bonding and adult attachment styles might play a key role in predicting group membership (i.e. patients with FM vs. HC) or otherwise if these variables could only predict the likelihood of having alexithymia.” Indeed, we wanted to test two different but related hypotheses. The first unexplored goal was to investigate if parental bonding and adult attachment styles might play a key role in predicting group membership (i.e. patients with FM vs. HC), while the second objective, based on some previous studies, was to assess if parental bonding and adult attachment styles could only predict the likelihood of having alexithymia, regardless of the presence of FM per se. 

For what concern the following sentence, ‘The present report represents a unique and different contribution, as we have never evaluated before attachment styles and the association with alexithymia in patients with FM’, it has to be noted that it was not present in the original version of the manuscript we submitted. After a preliminary revision made by the PLOS ONE editorial board, we were asked to indicate the differences and novelties of the present study with respect to our previous articles on FM (e.g. references [8; 14; 51-53]). However, in order to integrate better this paragraph with the twofold aim of our study, we slightly modified this part in the text (please, see lines 328-331). 

Regarding the second logistic regression analysis, we appreciate your suggestions to improve the methods section of our article. However, after careful consideration, we have thought that adding the variable ‘group’ (FM vs. HC) into the regression model, would not have been in line with the two above-outlined aims of our study. Indeed, we wanted to assess two different goals, the first one concerning the comparison between patients with FM and HC, whereas the second one relating to the comparison between alexithymic vs. non-alexithymic participants. Moreover, our group of FM patients, in line with the available evidence, reported significantly higher levels of alexithymia compared to the healthy women (35% of the FM patients vs. 8.4% of the HC scored above the TAS-20 cut-off); therefore, it is reasonable to assume that the variable ‘group’ might be a significant predictor in the final regression model, taking into account also the results of the first logistic analysis we performed (the only significant predictors in the final model were found to be the DIF and the DDF factors of the TAS-20).

The following suggestions and modifications should also be considered:

1) To delete the phrase “No previous study has been published yet, using the present dataset” from the measures section (line 123-124).

Thank you for your comment. We may not delete this sentence, as we were asked to insert this specification following the preliminary revision made by the PLOS ONE editorial board. 

2) It would be of interest to specify the Cronbach’s alphas obtained from the sample.

Thank you for your comment. We totally agree with you that it would have been more appropriate to report our own Cronbach’s alfa coefficients in the manuscript. However, we have some troubles in recovering the patients’ responses to the single items of the measures we employed. Indeed, we have administered all paper-and-pencil questionnaires and in the final dataset we have only the total scores for the different scales and subscales (we did not report the values for each item, as we used all validated and reliable instruments). The difficulties in recovering the patients’ responses are due to the impossibility to access to the questionnaires records as a consequence of the covid-19 health emergency. Indeed, the archives are located at the “Città della salute e della scienza” hospital of Turin, Italy. As a result, it would take months to retrieve all the necessary information for computing the Cronbach’s alfa coefficients on our data.

For this reason, we reported the Cronbach’s alfa coefficients from previous studies, which examined the psychometric properties of these instruments, in order to highlight their reliability and validity. 

3) There is no justification for the inclusion of the anxiety and depression measures as part of the study. The justification hasn’t been sufficiently well argued: “we decided to include also psychological distress variables into the model, considering the prominent role that these factors play in the symptomatology of patients with FM” (lines 197-198). Furthermore, the HADS was used as a measure for psychological distress, and it was later included as a measure for anxiety and depression.

Thank you for your observations. Following your insightful remark, we decided to remove anxiety and depression measures from our analyses. 

4) There is little clarity regarding the measure used to assess the sample’s educational level. Table 1 states “Educational level (years)”. Given that this is not a usual measure for educational level, what it actually means should be specified in the instruments section.

Thank you for your comment. We are aware that this is not a usual measure for the educational level. However, in order to obtain a continuous variable that could be easily used for data analyses, we asked our participants to indicate the total of how many years of education they achieved. In order to make this information clearer, we provided more details about the way educational level was assessed in the measures section (please, see ‘Sociodemographic and clinical information’ paragraph). 

5) Why was there no suggestion of a sample of healthy participants to match the fibromyalgia patients for age and educational level so to be able to ensure more homogeneity for these variables? Although no statistically significant differences were found, they are close to significance.

Thank you for your observation. We recruited healthy participants that were matched for demographic characteristics (i.e. age, gender, and educational level) to the FM patients. Following your suggestions, we specified better this information in the text (please, see ‘Participants and procedure’ and ‘Results’ sections). 

6) Were there any statistically significant differences in relation to marriage status or educational level between FM and HC? Chi-square analyses and p. values are missing in Table 1.

Thank you for your comment. Following your suggestions, we added these results in Table 1. 

7) Please indicate in the text (lines 231-234) the contrast statistic (chi-square) and the p. values for the differences in the proportion of alexithymia between the FM and HC groups.

Thank you for your comment. Following your suggestion, we added these results in the text. 

8) In the discussion, there is no need to comment on the association between parenting styles and other chronic diseases which are not fibromyalgia (lines 367-374). There should be further discussion regarding the results in relation to fibromyalgia populations.

Thank you for your observation. We included also studies carried out in different chronic pain populations, as only a limited number of studies is available in patients with FM (Gil et al., 2008; Peñacoba et al., 2018). Considering the sparse results on the topic, the first aim of the present study was exactly to shed light on the association between attachment variables and FM. 

9) In relation to the comment on lines 392-393 “Contrary to our expectations, neither parental bonding nor attachment styles significantly predicted group membership (i.e. patients with FM vs. HC)”, further discussion of this result is required. It is possible that having introduced symptoms (anxiety and depression) as predictive variables could have concealed the role that other variables of interest might have played. The method should be adapted, in particular the statistical analyses, for this limitation, and should include the use of symptom variables as possible modulators and not as predictors.

Thank you for your comments. Following your suggestions, we removed anxiety and depression measures (i.e. HADS A-D) from our analyses. However, the results of the first logistic regression we performed only slightly changed after removing the HADS A and D from the analysis. Indeed, once again the only significant predictor in the final model was found to be alexithymia (in this case both DIF and DDF factors of the TAS-20). 

10) In Model 2 (line 272) “In Model 2, the TAS-20 total and subscale scores were entered as predictors”, only dimensions and not total TAS-20 scores should have been included so to avoid co-linearity problems that could affect results. In fact, one of the dimensions included in the total TAS-20 (“externally oriented thinking”) has not been shown to be an explanatory variable in this study, nor in any other previous studies either.

We appreciate this remark being pointed out to us. Following your suggestion, we removed the TAS-20 total score from the first logistic regression analysis (please, see Table 3 for details).

---

## [Editor Report · Decision Letter 2]

26 Mar 2020

PONE-D-20-01350R2

Attachment style and parental bonding: relationships with fibromyalgia and alexithymia

PLOS ONE

Dear Dr Romeo,

Thank you for submitting your manuscript to PLOS ONE. After careful consideration, we feel that it has merit but does not fully meet PLOS ONE’s publication criteria as it currently stands. Therefore, we invite you to submit a revised version of the manuscript that addresses the points raised during the review process.

We would appreciate receiving your revised manuscript by May 10 2020 11:59PM. To enhance the reproducibility of your results, we recommend that if applicable you deposit your laboratory protocols in protocols.io, where a protocol can be assigned its own identifier (DOI) such that it can be cited independently in the future. For instructions see: http://journals.plos.org/plosone/s/submission-guidelines#loc-laboratory-protocols

We look forward to receiving your revised manuscript.

Kind regards,

Juan V. Luciano, Ph.D.

Academic Editor

PLOS ONE

Additional Editor Comments (if provided):

Dear authors,

I think you have correctly addressed all reviewers' concerns. Congratulations.

Notwithstanding, I encourage you to more deeply elaborate this idea in your discussion of results:

"First, when working with patients with FM, clinical attention should be paid to not only the management of pain symptoms but also impairments in affect regulation and attachment dynamics".

Your conclusion might have some link with the results of this recently published pilot RCT: Cost-Utility of Attachment-Based Compassion Therapy (ABCT) for Fibromyalgia Compared to Relaxation: A Pilot Randomized Controlled Trial - https://www.ncbi.nlm.nih.gov/pubmed/32156065

Sincerely,

Dr. Luciano

---

## [Author Response · Author response to Decision Letter 2]

27 Mar 2020

Turin, 25 March 2020

Ref: PONE-D-20-01350R1

Title: Attachment style and parental bonding: relationships with fibromyalgia and alexithymia

Journal: PLOS ONE

Dear Dr. Juan V. Luciano,

many thanks for the opportunity to resubmit the above manuscript to “PlOS ONE” journal. We are very grateful for the thorough, insightful reviews and we have modified the paper taking into account all these suggestions. Please, you can find below a point-by-point response to all the reviewers’ comments, in Italic the referee comments; in bold the authors’ responses. Modifications in the text have been highlighted in yellow. 

Waiting for your gentle reply,

Yours Sincerely 

Annunziata Romeo

 

Editor’s Comments

Dear Dr Romeo,

Thank you for submitting your manuscript to PLOS ONE. After careful consideration, we feel that it has merit but does not fully meet PLOS ONE’s publication criteria as it currently stands. Therefore, we invite you to submit a revised version of the manuscript that addresses the points raised during the review process.

Thank you for the opportunity to consider our manuscript for publication.

Reviewers’ Comments to Authors

Reviewer 1

Introduction.

Two recently published studies reported a higher prevalence of alexithymia in people with Fibromyalgia (48%-64%) than those reported in the introduction. It would be convenient to update this information.

References:

1. Front Psychol. 2019 Jul 31; 10: 1735. DOI: 10.3389/fpsyg.2019.01735.

2. Clin Exp Rheumatol 2019 Oct 9[Online ahead of print]

Thank you for your comments. Following your suggestions, we updated the prevalence of alexithymia in patients with Fibromyalgia (FM), according to this new evidence. 

Methods

1) A medical expert in the field made the diagnosis of Fibromyalgia. However, it is not clear if the doctor used any classification criteria (for example, ACR criteria 1990, 2010 or 2016) to unify the standards in the diagnosis in the sample. I consider it appropriate to clarify in the methods section.

Thank you for your comment. For the diagnosis of FM, the rheumatologists used the ACR criteria of 2010. Following your suggestion, we added this information in the ‘Participants and procedure’ section. 

2) Something to complement the description of the sample is if the center where the investigation was carried out, any patient with pain can go to the center (Primary Healthcare), or for their attention, the evaluation and referral of another doctor are necessary (Secondary / Tertiary Health Care).

Thank you for your observation. The usual practice for our center requires that firstly the general practitioner refers the patients to the rheumatologist, so that he/she can ascertain whether the diagnosis of FM may be done. After the diagnosis of FM has been confirmed, the rheumatologist usually refers the patients to the psychiatrist and psychologist, who in turn provide patients with the pharmacological treatment and the psychological support. 

3) There is duplicate information; for example, page 5, lines 106 and 112, repeat the time of inclusion of patients. The exclusion criteria are the same for both groups, with the exception that in the control group, patients with rheumatic diseases or chronic pain are excluded. I recommend presenting them in the same paragraph.

Thank you for your comments. Following your suggestions, we made the required changes to the ‘Participants and procedure’ section. 

Results

1) In the manuscript, there is any inconsistency between the number of people with fibromyalgia included. In the methods section, one hundred and seven patients report. In the headings of Table 1 and Table 2, 100 patients are reported. However, the sum of the variable's marital status and occupation includes only 99 patients. This information should be reviewed.

We appreciate this remark being pointed out to us. A total sample of 100 patients with FM was recruited for the present study. However, regarding the marital status, there has been a missing value and so the sum is 99 patients. For what concern, instead, both the ‘Participants and procedure’ section and the occupation, an error has occurred during the transcription of the data. We have now corrected the mistake in both parts. 

2) There is information that suggests gender differences in the prevalence of alexithymia. Because of the higher frequency of fibromyalgia in women, some studies only include women. The manuscript does not present the percentage of women involved in each of the groups. This information should be clarified.

Thank you for your observation. As indicated in the ‘Participants and procedure’ section, both the FM and HC groups consist entirely of women. Therefore, no percentage for gender has been reported. 

3) Page 11, line 220. Consider changing the word "composite" to "total score" or "global score." Which better reflects the score of the 20 items on the TAS-20 scale.

Thank you for your comment. Following your suggestion, we changed the word ‘composite’ to ‘total score’ for the TAS-20. 

Conclusions

In the conclusions, I suggest removing "In spite of these limitations".

Thank you for your observation. Following your suggestion, we removed ‘In spite of these limitations’ at the beginning of the conclusions. 

Reviewer: 2

Comments to the Author

This article focuses on a particular area of interest, specifically “to examine the associations between attachment styles, parental bonding, and alexithymia among patients with fibromyalgia (FM) and healthy controls (HC)” (lines 22-24), nevertheless the method employed makes it look as if it were two independent studies with no apparent link, as if they weren’t connected enough to be part of the same manuscript: a) to analyze predictive variables for fibromyalgia, b) to analyze predictive variables for alexithymia.

In particular, the manuscript presents two clearly differentiated analyses. One of them, directly associated to the aim of the manuscript, focuses on the differences between alexithymia, attachment, parental bonding and anxiety and depression in FM and HC by means of bivariate analyses and logistic regression. The other, with no association with the main aim, but having used the same method, uses the same analyses, in this case to analyze the differences between participants with and without alexithymia (with the total sample). There is a need to justify this analysis within the general aim of the study. In this context, and as the authors have pointed out (“the present report represents a unique and different contribution, as we have never evaluated before attachment styles and the association with alexithymia in patients with FM” (lines 349-351), the novelty of the manuscript is lost as it actually tackles two independent aims that have been widely reported in previous literature.

The methods should have been adapted, in particular the statistical analyses, so as to put both together (a and b). One possibility could be to include the condition (FM/HC) as a predictive or moderating variable (depending on which aim is being pursued) in the second regression analysis (alexithymia vs no alexithymia).

Thank you for your comments and appreciation. 

The present study had a twofold aim, as we reported in the following lines of the introductive section: “The present study aimed to examine deeply the associations between attachment styles, parental bonding, and alexithymia among patients with FM and healthy controls (HC). Particularly, we aimed to discern if parental bonding and adult attachment styles might play a key role in predicting group membership (i.e. patients with FM vs. HC) or otherwise if these variables could only predict the likelihood of having alexithymia.” Indeed, we wanted to test two different but related hypotheses. The first unexplored goal was to investigate if parental bonding and adult attachment styles might play a key role in predicting group membership (i.e. patients with FM vs. HC), while the second objective, based on some previous studies, was to assess if parental bonding and adult attachment styles could only predict the likelihood of having alexithymia, regardless of the presence of FM per se. 

For what concern the following sentence, ‘The present report represents a unique and different contribution, as we have never evaluated before attachment styles and the association with alexithymia in patients with FM’, it has to be noted that it was not present in the original version of the manuscript we submitted. After a preliminary revision made by the PLOS ONE editorial board, we were asked to indicate the differences and novelties of the present study with respect to our previous articles on FM (e.g. references [8; 14; 51-53]). However, in order to integrate better this paragraph with the twofold aim of our study, we slightly modified this part in the text (please, see lines 328-331). 

Regarding the second logistic regression analysis, we appreciate your suggestions to improve the methods section of our article. However, after careful consideration, we have thought that adding the variable ‘group’ (FM vs. HC) into the regression model, would not have been in line with the two above-outlined aims of our study. Indeed, we wanted to assess two different goals, the first one concerning the comparison between patients with FM and HC, whereas the second one relating to the comparison between alexithymic vs. non-alexithymic participants. Moreover, our group of FM patients, in line with the available evidence, reported significantly higher levels of alexithymia compared to the healthy women (35% of the FM patients vs. 8.4% of the HC scored above the TAS-20 cut-off); therefore, it is reasonable to assume that the variable ‘group’ might be a significant predictor in the final regression model, taking into account also the results of the first logistic analysis we performed (the only significant predictors in the final model were found to be the DIF and the DDF factors of the TAS-20).

The following suggestions and modifications should also be considered:

1) To delete the phrase “No previous study has been published yet, using the present dataset” from the measures section (line 123-124).

Thank you for your comment. We may not delete this sentence, as we were asked to insert this specification following the preliminary revision made by the PLOS ONE editorial board. 

2) It would be of interest to specify the Cronbach’s alphas obtained from the sample.

Thank you for your comment. We totally agree with you that it would have been more appropriate to report our own Cronbach’s alfa coefficients in the manuscript. However, we have some troubles in recovering the patients’ responses to the single items of the measures we employed. Indeed, we have administered all paper-and-pencil questionnaires and in the final dataset we have only the total scores for the different scales and subscales (we did not report the values for each item, as we used all validated and reliable instruments). The difficulties in recovering the patients’ responses are due to the impossibility to access to the questionnaires records as a consequence of the covid-19 health emergency. Indeed, the archives are located at the “Città della salute e della scienza” hospital of Turin, Italy. As a result, it would take months to retrieve all the necessary information for computing the Cronbach’s alfa coefficients on our data.

For this reason, we reported the Cronbach’s alfa coefficients from previous studies, which examined the psychometric properties of these instruments, in order to highlight their reliability and validity. 

3) There is no justification for the inclusion of the anxiety and depression measures as part of the study. The justification hasn’t been sufficiently well argued: “we decided to include also psychological distress variables into the model, considering the prominent role that these factors play in the symptomatology of patients with FM” (lines 197-198). Furthermore, the HADS was used as a measure for psychological distress, and it was later included as a measure for anxiety and depression.

Thank you for your observations. Following your insightful remark, we decided to remove anxiety and depression measures from our analyses. 

4) There is little clarity regarding the measure used to assess the sample’s educational level. Table 1 states “Educational level (years)”. Given that this is not a usual measure for educational level, what it actually means should be specified in the instruments section.

Thank you for your comment. We are aware that this is not a usual measure for the educational level. However, in order to obtain a continuous variable that could be easily used for data analyses, we asked our participants to indicate the total of how many years of education they achieved. In order to make this information clearer, we provided more details about the way educational level was assessed in the measures section (please, see ‘Sociodemographic and clinical information’ paragraph). 

5) Why was there no suggestion of a sample of healthy participants to match the fibromyalgia patients for age and educational level so to be able to ensure more homogeneity for these variables? Although no statistically significant differences were found, they are close to significance.

Thank you for your observation. We recruited healthy participants that were matched for demographic characteristics (i.e. age, gender, and educational level) to the FM patients. Following your suggestions, we specified better this information in the text (please, see ‘Participants and procedure’ and ‘Results’ sections). 

6) Were there any statistically significant differences in relation to marriage status or educational level between FM and HC? Chi-square analyses and p. values are missing in Table 1.

Thank you for your comment. Following your suggestions, we added these results in Table 1. 

7) Please indicate in the text (lines 231-234) the contrast statistic (chi-square) and the p. values for the differences in the proportion of alexithymia between the FM and HC groups.

Thank you for your comment. Following your suggestion, we added these results in the text. 

8) In the discussion, there is no need to comment on the association between parenting styles and other chronic diseases which are not fibromyalgia (lines 367-374). There should be further discussion regarding the results in relation to fibromyalgia populations.

Thank you for your observation. We included also studies carried out in different chronic pain populations, as only a limited number of studies is available in patients with FM (Gil et al., 2008; Peñacoba et al., 2018). Considering the sparse results on the topic, the first aim of the present study was exactly to shed light on the association between attachment variables and FM. 

9) In relation to the comment on lines 392-393 “Contrary to our expectations, neither parental bonding nor attachment styles significantly predicted group membership (i.e. patients with FM vs. HC)”, further discussion of this result is required. It is possible that having introduced symptoms (anxiety and depression) as predictive variables could have concealed the role that other variables of interest might have played. The method should be adapted, in particular the statistical analyses, for this limitation, and should include the use of symptom variables as possible modulators and not as predictors.

Thank you for your comments. Following your suggestions, we removed anxiety and depression measures (i.e. HADS A-D) from our analyses. However, the results of the first logistic regression we performed only slightly changed after removing the HADS A and D from the analysis. Indeed, once again the only significant predictor in the final model was found to be alexithymia (in this case both DIF and DDF factors of the TAS-20). 

10) In Model 2 (line 272) “In Model 2, the TAS-20 total and subscale scores were entered as predictors”, only dimensions and not total TAS-20 scores should have been included so to avoid co-linearity problems that could affect results. In fact, one of the dimensions included in the total TAS-20 (“externally oriented thinking”) has not been shown to be an explanatory variable in this study, nor in any other previous studies either.

We appreciate this remark being pointed out to us. Following your suggestion, we removed the TAS-20 total score from the first logistic regression analysis (please, see Table 3 for details).

---

## [Editor Report · Decision Letter 3]

30 Mar 2020

Attachment style and parental bonding: relationships with fibromyalgia and alexithymia

PONE-D-20-01350R3

Dear Dr. Romeo,

We are pleased to inform you that your manuscript has been judged scientifically suitable for publication and will be formally accepted for publication once it complies with all outstanding technical requirements.

With kind regards,

Juan V. Luciano, Ph.D.

Academic Editor

PLOS ONE
---

## [Editor Report · Acceptance letter]

31 Mar 2020

PONE-D-20-01350R3 

Attachment style and parental bonding: relationships with fibromyalgia and alexithymia 

Dear Dr. Romeo:

I am pleased to inform you that your manuscript has been deemed suitable for publication in PLOS ONE. Congratulations! Your manuscript is now with our production department. 

With kind regards,

on behalf of

Dr. Juan V. Luciano 

Academic Editor

PLOS ONE